# Moving Evidence into Practice by Advanced Practice Nurses in Hospitalization Wards. Protocol for a Multicentre Quasi-Experimental Study in Three Hospitals in Spain

**DOI:** 10.3390/ijerph17103473

**Published:** 2020-05-16

**Authors:** Sandra Pol-Castañeda, Miguel Ángel Rodríguez-Calero, Concepción Zaforteza-Lallemand, Carlos Javier Villafáfila-Gomila, Ian Blanco-Mavillard, Francisco Ferrer-Cruz, Joan De Pedro-Gómez

**Affiliations:** 1Hospital Son Llàtzer, 07198 Palma, Balearic Islands, Spain; sandrapolcas@gmail.com; 2Department of Nursing and Physiotherapy, University of the Balearic Islands, 07122 Palma, Balearic Islands, Spain; mianrodriguez@hmanacor.org (M.Á.R.-C.); depedro@uib.es (J.D.P.-G.); 3CurES Research Group, Balearic Islands Health Research Institute-IdISIBa, 07120 Palma, Balearic Islands, Spain; concha.zaforteza@hcin.es (C.Z.-L.); cvillafafila@ibsalut.es (C.J.V.-G.); 4Health Service of the Balearic Islands, 07003 Palma, Balearic Islands, Spain; 5Hospital de Inca, 07300 Inca, Balearic Islands, Spain; francisco.ferrer@hcin.es; 6Hospital Manacor, Quality, Teaching and Research Unit, Cra. de Manacor-Alcudia s/n, 07500 Manacor, Spain

**Keywords:** evidence-based practice, evidence-based nursing, advanced practice nursing, implementation science, organizational culture

## Abstract

Evidence-based practice (EBP) combined with quality of care improves patient outcomes. However, there are still difficulties for its implementation in daily clinical practice. This project aims to evaluate the impact of the incorporation of the Advanced Practice Nurse (APN) role on the implementation of EBP at three levels: context, nurses’ perceptions, and clinical outcomes. Mixed-methods study in two phases is proposed. Phase 1: a quasi-experimental design where five APNs are included in five hospitalization wards that are compared with another five similar wards without APNs. Variables from Practice-Environment-Scale-Nursing-Work-Index, Health-Science-Evidence-Based-Practice-Questionnaire, and Advanced-Practice-Nursing-Competency-Assessment-Instrument are used. Clinical outcomes are followed-up with monthly. A descriptive and exploratory analysis is performed. Phase 2: an exploratory qualitative design through focus groups at the intervention wards after one year of APNs implementation. Explicative data are gathered to explain the progression of change and how actors perceive and attribute triggers, barriers, and facilitators for change. An inductive thematic analysis is performed. The inclusion of APN in hospitalization context is insufficiently studied. It is hoped that these figures provide solutions to the multiple barriers in the development of EBP in these sceneries and contribute to resolve the gap between research results and healthcare practice.

## 1. Introduction

Evidence-based clinical practice (EBCP) has become an essential tool for the improvement of decision-making in health care [1,2,3]. Despite the widespread acceptance of this concept as a paradigm on which to base decisions in daily clinical practice [4], nearly 35% of these decisions do not conform to the recommendations offered by the best scientific evidence [5]. Multiple factors hinder the incorporation of best evidence into clinical practice [6,7,8], being an arduous and multifactorial task [9,10]. The volume of evidence and its variability is growing considerably, making it unmanageable for the clinician [11]. In addition, there is a delay in the implementation of research results caused by a resistance to change and the perception of clinical judgment as the core element in decision-making [12]. These facts may reduce the credibility of clinical practice guidelines (CPG) [13] and therefore increase the difficulty of their implementation, causing a suboptimal adherence to recommendations [13].

Health care systems have focused on minimising the variability of clinical practice [14]. Research agencies have conducted strategies worldwide for the effective transference of knowledge [15]. However, few proposals have taken into account the context and the organizational climate as relevant factors for change [16,17] and improvement of clinical outcomes [14,18]. The theory of planned behaviour (TPB) [19] and the Promoting Action on Research Implementation in Health Services (PARIHS) framework [20,21,22] are theoretical approaches underpinning the organisational change and facilitating the adherence to clinical practice guideline recommendations. In order to implement this theoretical model, attention should be focused on the facilitator role of the best evidence played by the Advanced Practice Nurse (APN) [23,24]. This approach would address two attention levels: individual, seeking to modify the work patterns of health care personnel; and organisational to facilitate this change.

In both of these approaches, a facilitating element can be characterised as an agent of change that predominantly uses a participatory problem-solving method rather than prescribing and directing a system of actions [25,26]. One of the solutions that is gaining the most popularity worldwide is to recruit APNs, whose high-level postgraduate training and advanced knowledge and skills make them very appropriate for this task, to raise the level of EBCP within a given environment. In this respect, studies performed in different contexts have acknowledged the value of APNs as agents of change [27,28], capable of influencing their work environment through the transfer of evidence-based knowledge and thus producing disruptive innovation [29]. These nurses are especially well qualified for this task because their professional practice is based on competencies related to research and EBCP, acting as leaders in their context regarding the adoption of updated clinical practices [30]. It has been shown that optimising nurses’ contributions to health care by expanding their capabilities and scope of practice is an effective strategy for improving the overall quality of health services [31,32,33]. The work performed by APNs is safe, effective, and well received by users [34].

This project represents an initial experience to explore possible care improvements for the future application of the functions of the advanced practice hospitalization nurse (APHN) in our health system. For this intent, we propose APNs be incorporated into hospital wards as a new approach to the role of advanced practice nursing in our country. By doing so, we assess the value of this new role—one that puts into practice the clinical leadership—and the process of incorporating that role within the team from the perspective of the nurses involved. Moreover, the elements of the PARIHS framework are taken into account as a facilitating element of behavioural change based on the TPB for the implementation of EBCP within a favourable context supported by the health care agency.

Therefore, the main objective of this study is to evaluate the impact of the incorporation of the APHN role compared to hospital wards without APHN role. The specific objectives are to (1) evaluate the impact of APHNs on the EBCP environment (contextual and individual factors); (2) explore the bedside nurses and APHNs’ perceptions towards EBCP concerning the incorporation of the APHNs in their hospital ward; (3) determine the impact of APHN-directed interventions on clinical outcomes of inpatients; and (4) assess the competencies of the APHNs who take part in the project.

## 2. Materials and Methods

### 2.1. Design

This protocol describes an ongoing project, currently at the intervention stage, consisting of a mixed-methods study in two phases with a first quasi-experimental design and a second exploratory qualitative phase through focus groups (Figure 1).

### 2.2. Settings

Ten medical and surgical hospital wards were selected in three public hospitals managed by the Balearic Islands health service—a university hospital and two regional hospitals. Intervention and control groups were created, each consisting of three medical and two surgical wards.

### 2.3. Interventions

The nurse who develops the role of APHNs was replaced by another registered nurse who assumes the habitual assignment of patients. The replacement of the five nurses was covered by the Health Service of the Balearic Islands (IB-Salut) who gave financial support to the project in these terms. Two registered nurses were selected from each unit (one APHN and one support nurse) and, together with the ward’s supervisor, received specific training (Table 1) in line with the theoretical framework of the project and the competencies required [30]. All APHNs were monitored and advised by the research team via individual contact as necessary and at monthly meetings between the research team, the APHNs, the support nurses, and the supervisors. The support nurse (whose normal assignment of patients was maintained) and the supervisor assisted with, in conjunction with the APHN, the introduction of new work dynamics to improve the environmental outcomes within the ward. The APHNs designed specific actions to put into effect two clinical practice guidelines (CPG), adapted as appropriate to the specific context and its characteristics The selection of the CPGs was based on the following criteria: (1) the number of guidelines was limited so that the APHN-directed interventions could be designed and monitored within the study period; (2) CPG topics had to be standard and applicable to any of the intervention wards regardless of patient or team characteristics; (3) CPG topics had to be of high impact in inpatient health and aligned with the Health Service’s strategic map. Therefore, the selected guidelines were “Care and maintenance of vascular access to reduce complications” [35] and “Prevention and treatment of pressure ulcers” [36].

The APN group, together with the support nurses and supervisors, are in frequent contact through meetings, initially weekly and later monthly. Here, they can assess each other’s progress and, with the research team, share material, exchange ideas and experiences, and ensure that the research data are obtained correctly.

In the control wards, no intervention was performed to improve or facilitate the usual pattern of treatment. The indicators measured were the same as in the intervention wards, enabling us to compare the results obtained in each case.

### 2.4. Variables

In the quantitative phase, four main study variables are addressed via Spanish validated instruments (Appendix A):-Organisational traits, measured by the Practice Environment Scale Nursing Work Index (PES-NWI) [37].-The nurses’ individual level of EBCP, measured by the Health Science Evidence-Based Practice Questionnaire (HS-EBP) [38].-Health indicators of the patients admitted to the study wards, derived from the clinical practice guidelines implemented by the APHNs in their wards and obtained by monthly clinical audits (Table 2).-The APHNs’ competence, as measured by the Advanced Practice Nursing Competency Assessment Instrument (APNCAI) [39].

In the qualitative phase, after 1 year of the APHN’s implantation, the nurses’ perceptions are explored regarding the evidence-based clinical practice and the impact on the health results of the patients (Table 2).

### 2.5. Sample and Participants

The quantitative phase started with registered nurses working in the hospital wards included in the study in addition to the APHNs and their supervisors working in the intervention wards. The qualitative phase is carried out only with the nurses of the intervention group, the APHN, and the supervisors.

The sample size needed for the quantitative phase was calculated according to the difference of expected scores of the PES-NWI questionnaire on organizational traits (Appendix A). The calculation was performed for independent means, assuming an alpha risk of 0.05 and a beta risk of less than 0.2 in bilateral testing. Therefore, to detect a difference equal to or greater than four points, 70 subjects are needed in each of the study groups, assuming a standard deviation of eight and a loss rate of 10%. In the Health Service of the Balearic Islands, there are about 15 nurses per hospital unit, thus each study group corresponds to five hospital wards.

In the qualitative phase, six focus groups are carried out: one focus group with the APHNs and five focus groups with nurses, one in each of the five units where the figure of the APHN is implanted. Sampling for the focus groups of nurses is intentional, and a minimum of six and a maximum of 12 nurses are recruited in each of the units. The five APHNs and at least three to five support nurses are included in the APHN focus group.

### 2.6. Inclusion and Exclusion Criteria

Quantitative phase: the intervention wards were selected according to the baseline score obtained in the PES-NWI questionnaire, although no cut off points were established. Medical and surgical wards with a well-established leadership structure were included with the participation of each hospital’s nursing managers. Each intervention unit was paired with a control unit in the same hospital with similar characteristics and health indicators.

After the selection of hospital wards, project information meetings were held for the nursing teams, who were invited to participate. The nurses in each unit nominated one of their number, presenting the competencies needed to perform the role of the APHN, and another who provided support. Both participants had to be in a stable employment relationship in the unit. This selection was made from the nurses who applied to take part, according to their leadership qualities and APNCAI scores obtained (Appendix A). The supervisors of the wards were also part of the study subjects.

Qualitative phase: for the focus groups in the intervention groups, we select registered nurses who have worked throughout the entire intervention period. For the APHN focus group, the inclusion criteria is having been the APHN of the unit from the beginning of the project.

### 2.7. Data Collection

Quantitative data are compiled in three moments: before, during, and after the intervention. In January 2018, the PES-NWI data were obtained to select the study wards, as already stated. Subsequently, the HS-EBP and the APNCAI instruments were provided in the pre-intervention phase (April 2018). These three instruments (PES-NWI, HS-EBP, and APNCAI) were used again in the post-intervention phase, one year after the incorporation of the APHNs into the corresponding hospital wards (September 2019). The pre- and the post-phase data collection operations were performed by the research team. During the intervention itself, the data derived from the clinical indicators were collected by the APHNs, the support nurses, and the supervisors of the wards involved, who received appropriate training for this task, to standardise the technique. This data collection was carried out through audits based on direct observation of the patient’s condition and a review of clinical records, both in the control unit and in the intervention unit. The data were stored in a common data file for analysis.

Qualitative data will be collected in May 2020. The focus group registration data will be done through a double audio recording and subsequent transcription. It will be carried out in between October and December 2020, at the same hospital or in the place decided by the participants, which encourages their participation, tranquility, and comfort. Six focus group will be done with the APHNs under the same conditions. The duration of the focus groups is expected to be in between 45 min and one hour, approximately.

### 2.8. Data Analysis

Descriptive and exploratory analyses of the quantitative data will be performed, determining, among other parameters, the main indices of central tendency, dispersion, and shape. In addition, an analysis of missing values will be conducted. The association between variables will be determined according to the scale of their mean values. A stratified analysis will be carried out using the Mantel and Haenszel test, among others (stratifying according to the possible confounding variables). Bivariate analysis will be performed to detect significant associations between the main variables and sociodemographic, occupational, and complementary variables. Multivariate analysis will also be carried out using regression models. The mean values obtained will be compared by ANOVA or the nonparametric equivalent (Kruskal–Wallis test). The main variables will be crossed with the sociodemographic variables (age, gender), the occupational variables (hospital, unit, years of experience as a registered nurse), and the complementary variables (academic qualifications). These will be analysed as possible confounders and/or modifiers of the effects produced on the study variables. Finally, an exploratory analysis of lost data will be performed. The SPSS v.22 statistical software package ( SPSS Inc., Chicago, IL, USA) will be used for all these analyses.

For the qualitative phase, a realistic thematic analysis [40] of the focus groups transcriptions will be carried out. It will be used to generate a rich description of registered nurse and APHN views on how the change process evolved and impacted the unit and patients’ outcomes and how context jeopardized and/or facilitated the process. Two researchers will analyze the data across five phases: (1) through a first process if reading, initial themes and codes will be identified and grouped and emerging patters will be checked; (2) a second process of reading and re-reading, aimed at interpreting the themes, will be conducted; (3) at this point, the two researchers will share their interpretation and will reach consensus on the list of codes and sub-codes; (4) these codes will be rechecked through a third process of reading then pieces of text that exemplify the themes will be categorized under each code. Attention will be paid to new codes that may not have been identified earlier; (5) final patterns and links will be identified among themes and codes.

The “Atlas-Ti” tool will be used for the trees’ codification, and the results will be triangulated among other members of the research team. A review by the participants will also be requested (member checking).

### 2.9. Rigour and Validity

The randomization was not possible in this quasi-experimental study. Its strength lies in the controlled nature of the APN interventions and the continuous support offered in the form of permanent communication, training, and mentoring. The fact that the study is multicentre and was performed in very diverse contexts and at different levels from the type of patient to the type of organisation increases its external validity and means that a rich database of both quantitative and qualitative data will be obtained for analysis.

### 2.10. Ethics Approval and Consent to Participate

The study has been approved by the Balearic Islands Research Ethics Committee of the Health Department of the Balearic Islands (IB No. 3662/18 PI) on 27/06/2018 (committee No. 06/18).

The data collection process will be coded to ensure the anonymity of those taking part. Before completing the questionnaires and recording the necessary clinical parameters, informed consent will be obtained from all concerned, by Act 41/2002, of 14 November, which regulates patients’ autonomy and their rights and obligations in terms of general and clinical information. Throughout this study, the research team will follow the applicable provisions of the Declaration of Helsinki and those of Spanish legislation concerning clinical research, with particular regard to Act 14/2007, of 3 July, on biomedical research. All participants will sign an informed consent for the audio recording of the focus groups and will be informed that, under no circumstance, their opinions will be linked to their identity. The study data will be processed and stored by Regulation (EU) 2016/679 (RGPD) and the current legislation on data protection, thus ensuring confidentiality.

## 3. Discussion

The role of APNs is officially recognized in nine out of the 17 Regional Health Services in Spain, but this is the first time they have been incorporated into the Balearic Islands health system, where it is not regulated at present [41]. The presence of a nurse working specifically to facilitate the implementation of evidence-based practice according to contextual and individual characteristics of a medical ward could significantly enhance the organizational environment. Moreover, the introduction of such a figure could improve the attitudes of health personnel towards necessary changes and heighten the individual perceptions of nurses regarding these changes. The presence of APHNs is expected to reinforce evidence-based clinical practice and thus reduce variability in this regard, thereby achieving tangible improvements in the quality of patient care [28,42].

APHN-directed interventions are expected to improve patient health conditions reducing healthcare-related complications derived from peripheral intravenous cannulation (mainly phlebitis, infection, and obstruction) and preventing the appearance of pressure ulcers. In consequence, the incidence of these complications is expected to be lower in intervention wards. Key elements of previous interventions proven to be effective in this respect [43,44] will be deployed by the APHNs in other to mobilize and re-order nurses activities and attitudes in their respective contexts [24].

In addition, APHNs extended abilities are expected to mobilize available resources and undertake new activities or services to improve health-related variables among inpatients. Furthermore, we believe these interventions may improve the perception of evidence-based practice among nursing team members, which will surely improve the effectiveness of the activities they are carrying out [44].

A limitation of the present research may be that it will not detect the overall impact of the incorporation of APHNs into the Balearic Islands Health System immediately due to the duration of the study. We consider that it should be necessary to continue measuring these interventions after this period of study in order to reach a more complete knowledge about the overall impact in clinical outcomes. In addition, the possible existence of informal leadership profiles within the control wards should be taken into account, as this circumstance could produce changes or improvements in parallel to those promoted by the APHN. There is also the possibility of contagion, with the control wards benefiting from this intervention. As they are located within the same health centre, there may be information flows among the medical staff involved. Thus, if a given intervention proves to be effective, the nurses in the control unit could become aware of this, apply the corresponding changes, and obtain similar improvements in the variables. Finally, the limited number of guidelines and variables selected for our study will probably leave some potential effects out of the scope of this investigation that would require further research.

Depending on the findings obtained, and in the event that the organisations consider it advisable to retain the new role of APHNs in the system, further research evaluations may be continued to address the limitations identified above.

The present project allows us to evaluate the adequacy of the implementation of APHN role in hospitals of our context. This implementation would be a major event that could produce a significant qualitative advance in the current organizational structure of hospitals in the Balearic Islands, not only in the area of nursing but in general hospital management.

Finally, the study contributes to resolving the research-practice gap and how to ensure that clinical evidence becomes accepted in daily nursing practice.

## Figures and Tables

**Figure 1 ijerph-17-03473-f001:**
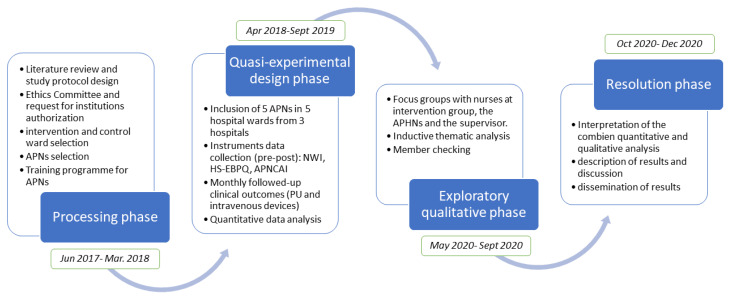
Phases of the study.

**Table 1 ijerph-17-03473-t001:** Training programme for Advance Practice Nurses.

COMPETENCIES	CONTENT	Theory of Planned Behaviour (TPB) FOCUS
Basic aspects	Concepts of Advanced Practice Nursing and Evidence-Based Clinical Practice.	Attitude, subjective norms
	State of the art in Spain and abroad.	
	Introduction to the theories and models of knowledge transfer.	
Instrumental training	Presentation of the Clinical Practice Guidelines to be used.	Perceived control
	Indicators, methods of data collection, and use.	
Methodology of health care research	Variability in clinical practice.	Attitude, perceived control
	Fundamentals of research and critical thinking.	
	Management of documentary sources.	
	Quantitative and qualitative research methods; meta-analysis and meta-synthesis.	
	Communication and scientific writing. Critical reading.	
	Research ethics.	
Clinical and professional leadership	Introduction to the concept of leadership.	Subjective norms, perceived control
	Language and human communication. Skills for efficient communication.	
	Development and implementation of innovations.	
	Managing networks of engagement.	
	The APN as an evidence-based consultant.	
	Acquiring skills for assertive communication.	
	Managing conflicting or negative situations.	
Nursing autonomy	Spanish legislation on nursing and health care.	Attitude, subjective norms
	Nurse prescribing.	
	Framework for APN performance.	
	Code of nursing ethics.	
	Nursing methodology and nursing care plans.	
	Costs of health services.	
	Clinical treatment plans.	
Interprofessional relations and mentoring	Interprofessional communication and problem-solving techniques.	Perceived control
	Clinical supervision based on mediation and mentoring.	
	Academic tutoring as an element fostering quality, engagement, and the acquisition of new knowledge and skills.	
Quality control	Advanced practice nursing as a means of reducing variability in clinical practice.	Attitude, subjective norms
	Theories and models of knowledge transfer as an essential element in the application of new research findings.	
	Innovation as a key element for change and improvement in clinical practice.	
	Implementation based on results and evaluation.	
Management of patient care	Organigram of the Balearic Islands Public Health System.	Perceived control
	Financing health care systems.	
	Nursing as the basis of the care process and the requirement for its continuity.	
	Patient scheduling and case management.	
	Supervising the results of health care and its impact on clinical management and interventions.	
	Training as the basis for clinical care for patients and their families.	
Promoting health care	Health policies, locally and internationally.	Attitude, Perceived control
	Techniques and strategies for promoting health.	
	Development and implementation of health promotion programmes for adolescents and adults.	
	Secondary and tertiary prevention for people with multiple or chronic health problems.	
	Promoting self-care within the family and/or providing support systems to facilitate their participation in health care.	
	Empowering the individual, groups and communities to adopt healthy lifestyles and self-care.	

**Table 2 ijerph-17-03473-t002:** Clinical indicators.

Indicators Derived from Clinical Practice Guidelines (CPG) for Pressure Ulcers (PU)				
	Indicator	Description	Definition	Data Collection Method
Structure indicators	Knowledge of the CPG	Awareness of the CPG within the unit	Number of nurses in the unit familiarised with the CPG × 100/number of nurses in the unit	By direct interview, performed annually.
	Material resource requirements	Wards presenting a material resource needs report	Dichotomous value. Does the unit present a needs report?	Audited by APNs, annually
Process indicators	Risk of PU	Assessment of PU risk in the target population, according to the Braden Scale	Number of patients at risk of PU according to the Braden scale × 100/number of hospitalised patients	By computer system and monthly APN audit
	Postural changes	Percentage of patients at risk of PU with an appropriate schedule of postural changes	Number of patients with scheduled postural changes × 100/number of patients at risk of PU	Monthly APN audit
	Pressure modification/Pressure relief support (PMS/PRS)	Percentage of patients at risk of PU and provided with PMS or PRS	Number of patients provided with a viscoelastic mattress × 100/number of patients at low, moderate and high risk of PU, according to the Braden scale	Monthly APN audit
Outcome indicators	Prevalence of PU	Percentage of patients with PU when the study was performed (or obtained monthly)	Number of patients with PU × 100/number of patients who met the criteria for inclusion in the study (hospitalised patients)	By computer system and monthly APN audit
	Prevalence of PU according to risk	Percentage of patients at risk of PU who presented PU when the study was performed	Number of patients with PU × 100/number of patients at risk during the study period	By computer system and monthly APN audit
	Incidence of PU	Percentage of patients initially without PU who developed PU during the study period	Number of patients initially free of PU who developed at least one PU during the study period × 100/cumulative number of patients during the study period who met the inclusion criteria	By computer system and monthly APN audit
	Incidence of PU according to risk	Percentage of patients at risk initially without PU who developed PU during the study period	Number of patients initially free of PU who developed at least one PU during the study period × 100/cumulative number of patients at risk (according to the Braden scale) during the study period	By computer system and monthly APN audit
Indicators derived from Clinical Practice Guidelines (CPG) for vascular access devices				
	Indicator	Description	Definition	Data collection method
Structure indicators	Knowledge of the CPG	Awareness of the CPG within the intervention unit	Number of nurses in the unit who are familiarised with the CPG × 100/number of nurses in the unit	By direct interview, performed annually.
Process indicators	Adherence to recommendations	Rate of adherence to the CPG recommendations on nursing practice regarding the location, calibre, and visualisation of the catheter	Number of catheters inserted in the antecubital fossa × 100/total number of catheters inserted in upper limbs;	Monthly APN audit
			Number of catheters inserted in lower limbs × 100/total number of catheters;	
			Number of catheters with the orifice visible to the naked eye × 100/total number of catheters	
	Record	Recording of nursing procedures with respect to location, calibre, and visualisation of the catheter	Number of catheters recorded as inserted in the antecubital fossa × 100/number of catheters recorded as inserted in upper limbs;	By computer system and monthly APN audit
			Number of catheters recorded as inserted in lower limbs × 100/total number of catheters recorded as inserted;	
			Number of catheters recorded as inserted, with the orifice visible to the naked eye, × 100/total number of catheters recorded as inserted	
Outcome indicators	Prevalence of adverse events	Prevalence of adverse events related to the peripheral catheter, presenting as phlebitis, extravasation, obstruction, infection, accidental withdrawal, malfunction, or purulent exudate	Number of adverse events/number of records × 100	By computer system and monthly APN audit
	Incidence of adverse events	Incidence of adverse events related to the peripheral catheter, presenting as phlebitis, extravasation, obstruction, infection, accidental withdrawal, malfunction, or purulent exudate	Number of adverse events since the APN entered the unit/number of patients hospitalised during the year × 100	By computer system and monthly APN audit
	Unnecessary catheters	Rate of unnecessary peripheral catheters	Number of unnecessary catheters/total number of catheters × 100	Monthly APN audit

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
