# Peer review of "Moving Evidence into Practice by Advanced Practice Nurses in Hospitalization Wards. Protocol for a Multicentre Quasi-Experimental Study in Three Hospitals in Spain"

_ijerph, 2020, doi:10.3390/ijerph17103473_

Round 1

Reviewer 1 Report

Thank you very much for the opportunity of revising this interesting paper.

I think that the movement from the evidence to the evidence-based practice in nursing is a huge challenge. To reach it and overcoming this theory-practice gap, this kind of research are needed. 

So, results, not protocols, about this kind of experiences, can set the good examples for others to follow.

Methodologically this protocol is well planned, and a mixed-method may make better and complementary results. But results are needed to assess the effectiveness of a project.

Author Response

Reviewer 1

Thank you very much for the opportunity of revising this interesting paper.

I think that the movement from the evidence to the evidence-based practice in nursing is a huge challenge. To reach it and overcoming this theory-practice gap, this kind of research is needed.

So, results, not protocols, about this kind of experience, can set good examples for others to follow.

Methodologically this protocol is well planned, and a mixed-method may make better and complementary results. But results are needed to assess the effectiveness of a project.

R: We are very grateful to the reviewer for the encouragement. Many thanks for the suggestion. The publication of this research protocol aims to provide the key elements that are part of the design of our study, which will be available to the scientific community, integrating quantitative and qualitative methodology. Rather than reporting research results, we intend to share methodological details of our study so as to make it reproducible in other contexts.

Reviewer 2 Report

I congratulate the authors for their important work, I firmly believe that it will bring great contributions to the field of Nursing.

I present some suggestions and proposals for adjustments that should be taken into account.

The abstract is out of the journal's standards. Suggest that it be more objective and present the most relevant points of the study.

The purpose of the study is not mentioned at the end of the introduction. Please insert.

About methods: The limitations of the study are confusing. Detail the limitations and potential of the research. How will the qualitative analysis be carried out? Detail point to point. Please provide more details on ethical aspects, for example: respect for the Helsinki protocol.

Discussion: Please provide more details about the expected results. Better justify the importance of carrying out this study.

Author Response

Reviewer 2

I congratulate the authors for their important work, I firmly believe that it will bring great contributions to the field of Nursing.

I present some suggestions and proposals for adjustments that should be taken into account.

The abstract is out of the journal's standards. Suggest that it be more objective and present the most relevant points of the study.

R: We are grateful to the reviewer for these suggestions. We have modified the abstract adapting it to the journal’s standards.

The purpose of the study is not mentioned at the end of the introduction. Please insert.

R: We have now moved the purpose of the study to the end of the introduction. 

About methods: The limitations of the study are confusing. Detail the limitations and potential of the research.

R: Thank you for these questions. We have clarified now and written in full.A limitation of the present research may be that it will not detect the overall impact of the incorporation of APHNs into the Balearic Islands Health System immediately, due to the duration of the study. We consider that it should be necessary to continue measuring these interventions after this period of study, in order to reach a more complete knowledge about the real impact of clinical outcomes.

How will the qualitative analysis be carried out? Detail point to point.

R: Thank you for this suggestion. We have now amended to ‘For the qualitative phase, a realistic thematic analysis[44] of the focus groups transcriptions will be carried out. It will be used to generate a rich description of RN and APHN views on how the change process evolved, impacted the unit and patients’ outcomes, and how context jeopardized and/or facilitated the process. Two researchers will analyze the data across 5 phases. 1) Through a first process if reading, initial themes and codes will be identified, grouped and emerging patters will be checked; 2) A second process of reading and re-reading, aimed at interpreting the themes, will be conducted; 3) At this point, the two researchers will share their interpretation and will reach consensus on the list of codes and sub-codes; 4) These codes will be rechecked through a third process of reading, then pieces of text which exemplify the themes will be categorized under each code. Attention will be paid to new codes that may not have been identified earlier; 5) Final patterns and links will be identified among themes and codes.

Please provide more details on ethical aspects, for example: respect for the Helsinki protocol.

R: We have now added more details to Ethics approval and consent to participate section.

Discussion: Please provide more details about the expected results. Better justify the importance of carrying out this study.

R: Many thanks for the question. We have now added more details to the discussion. ‘APHN-directed interventions are expected to improve patient health conditions reducing healthcare-related complications derived from peripheral intravenous cannulation (mainly phlebitis, infection, and obstruction) and preventing the appearance of pressure ulcers. In consequence, the incidence of these complications is expected to be lower in intervention wards. Key elements of previous interventions proved to be effective in this respect[46,47] will be deployed by the APHNs in other to mobilize and re-order nurses activities and attitudes in their respective contexts[25].

In addition, APHNs extended abilities are expected to mobilize available resources and undertake new activities or services to improve health-related variables among inpatients. Furthermore, we believe these interventions may improve the perception of evidence-based practice among nursing team members which will surely improve the effectiveness of the activities they are carrying out[47].

Reviewer 3 Report

This topic of strategies to translate research into practice and strategies to support high level evidence based clinical practice for all hospitalized patients is timely and important.

Need to clearly identify the educational and professional criteria for the APN, APHN, supervisory nurse, and nurse

Need to clearly identify sources for the Clinical Practice Guidelines (CPG) and the Evidence Based Clinical Practice (EBCP) principles/standards as evidence of current research and standards of care.

Should explain why the focus on quality is limited to CVC and pressure ulcer prevention

Concern: 

Use of editorial, opinion, resources such as reference #11 and #12 can reduce the validity of the significance of this research

Recommend removal of old items published greater than 10 years ago as references unless the discussion in the manuscript clearly uses them as source of old perspectives t contrast with movement to evidence based clinical practice. 

Unclear of purpose for reference #13, Survivorship Guidelines, is used since this is an old tool for specialist in oncology to provide a document about the ongoing health management for survivors of cancer as a resource for the patient and their primary care physician once the oncologist no longer manages the patient's health.

Grammar issues

line 134, 218

Spelling:  line 218  'randomisation'  change to 'randomization'

Line 79: unclear aim statement: “Explore the nurses’ individual perceptions….. which nurses—bedside or APN

Author Response

Reviewer 3

This topic of strategies to translate research into practice and strategies to support high level evidence based clinical practice for all hospitalized patients is timely and important.

Need to clearly identify the educational and professional criteria for the APN, APHN, supervisory nurse, and nurse

R: We appreciate the review, thank you. We have now clarified the inclusion and exclusion criteria.

Need to clearly identify sources for the Clinical Practice Guidelines (CPG) and the Evidence Based Clinical Practice (EBCP) principles/standards as evidence of current research and standards of care. Should explain why the focus on quality is limited to CVC and pressure ulcer prevention

R: Thank you for suggestions. We have now clarified the intervention section. We have also included this issue in the limitations section.

Concern:

Use of editorial, opinion, resources such as reference #11 and #12 can reduce the validity of the significance of this research

R: We agree with your suggestion and we have changed reference #11 to Greenhalgh, T.; Howick, J.; Maskrey, N. Evidence based medicine: a movement in crisis? BMJ 2014, 348, g3725–g3725.

Recommend removal of old items published greater than 10 years ago as references unless the discussion in the manuscript clearly uses them as source of old perspectives t contrast with movement to evidence based clinical practice.

R: Thank you for the suggestion. We have reviewed all the manuscript removing references published before 10 years ago.

Unclear of purpose for reference #13, Survivorship Guidelines, is used since this is an old tool for specialist in oncology to provide a document about the ongoing health management for survivors of cancer as a resource for the patient and their primary care physician once the oncologist no longer manages the patient's health.

R: We agree with your suggestion and we have removed reference #13

Grammar issues

line 134, 218

Spelling:  line 218  'randomisation'  change to 'randomization'

Line 79: unclear aim statement: “Explore the nurses’ individual perceptions….. which nurses—bedside or APN

R: Thank you, we have reviewed the manuscript for English spelling and amended whenever necessary.

Round 2

Reviewer 1 Report

Thank you very much for modifications done in the protocol.

I am aware of the good intentions of the authors presenting a protocol with all possible details. In this way it could be replicated in other health centers.

However, this same information could be presented in an article in which the results of the protocol were also provided. Thus, in addition, its effectiveness would be shown. Evidence must be provided in scientific research. And one way of doing it to improve the nursing profession and the scientific nursing literature is to provide methodologies and results on the research carried out.

I invite the authors to share the results they obtain with this project. It is a very interesting project, which can improve the quality of care for people in the public health system, and with a high probability of success.

Author Response

R: We are grateful to the reviewer for these suggestions. We also thank you for understanding our intention and the benefits of the publication of this study protocol.

We consider such a publication will be valuable for other researchers interested in developing research projects aimed to facilitate knowledge transference, being an essential element of transparency. As you already know, the publication of study protocols reduces the possibility of redesigning or adapting analysis, and decreasing the discrepancies between the protocol accepted by the ethics committees and the final publication of results. We consider that protocol study guarantees the methodological rigour and validity of the results, regardless of the potential interests of the authors.

Additionally, this format offers an opportunity for us to explain with detail all methodological aspects that would be more limited in a full results report considering the well-known limitation of manuscript extension in scientific journals.

As we explained, we believe that writing a protocol paper also allows us to define the variables with accuracy and detail each part of the methodology in a way that it can replicate and adapt by other research groups.

Reviewer 2 Report

Dear authors, thank you for accepting all suggestions. The article needs to undergo a revision in the sense of the English language.
Graciously;

Author Response

R: Thank you for the suggestion. We have revised the English language.